# A scalable molecule-based magnetic thin film for spin-thermoelectric energy conversion

Inseon Oh[1], Jungmin Park[2], Daeseong Choe [1], Junhyeon Jo [1], Hyeonjung Jeong[1], Mi-Jin Jin[1,3], Younghun Jo[2], Joonki Suh[1], Byoung-Chul Min [4] & Jung-Woo Yoo [1✉]

Spin thermoelectrics, an emerging thermoelectric technology, offers energy harvesting from waste heat with potential advantages of scalability and energy conversion efficiency, thanks to orthogonal paths for heat and charge flow. However, magnetic insulators previously used for spin thermoelectrics pose challenges for scale-up due to high temperature processing and difficulty in large-area deposition. Here, we introduce a molecule-based magnetic film for spin thermoelectric applications because it entails versatile synthetic routes in addition to weak spin-lattice interaction and low thermal conductivity. Thin films of $Cr^{II}[Cr^{III}(CN)_6]$, Prussian blue analogue, electrochemically deposited on Cr electrodes at room temperature show effective spin thermoelectricity. Moreover, the ferromagnetic resonance studies exhibit an extremely low Gilbert damping constant $\sim(2.4 \pm 0.67) \times 10^{-4}$, indicating low loss of heat-generated magnons. The demonstrated STE applications of a new class of magnet will pave the way for versatile recycling of ubiquitous waste heat.

[1] Department of Materials Science and Engineering, Ulsan National Institute of Science and Technology, Ulsan, Korea. [2] Division of Scientific Instrumentation & Management, Center for Scientific Instrumentation, Korea Basic Science Institute, Daejeon, Korea. [3] Soft Chemical Materials Research Center for Organic-Inorganic Multi-dimensional Structures, Dankook University, Yongin, Korea. [4] Center for Spintronics, Korea Institute of Science and Technology, Seoul, Korea. ✉email: jwyoo@unist.ac.kr

Thermoelectric (TE) technologies offer energy harvesting from the most common form of energy "heat." A TE device is solid state, highly reliable, environmentally friendly, and compact in comparison with other energy conversion generators, desirable features for collecting omnipresent heat energy. Spin TE (STE), an emerging generation of TE technologies, converts heat into electricity via the spin Seebeck effect (SSE) using a combination of two layers[1–3]. One is a magnetic insulator, where the temperature gradient induces the propagation of thermally exited spin wave, magnon[4]. The other layer is a nonmagnetic heavy metal, such as, Pt, W, and Ta with strong spin–orbit coupling for the effective spin–charge conversion via the inverse spin Hall effect (ISHE)[5,6]. An applied vertical heat flux in the ferromagnet/heavy-metal bilayer produces longitudinal electrical power as illustrated in Fig. 1a. The bilayer architecture of an STE device has several advantages over traditional TE modules[3]. For example, in conventional TEs the figure of merit ($ZT = S^2\sigma/\kappa$, where $S$, $\sigma$, and $\kappa$ are Seebeck coefficient, electrical conductivity, and thermal conductivity, respectively) of the energy conversion is strictly limited by the trade-off relationship between $S$ and $\sigma$, as well as the Wiedemann–Franz law[7]. In STE, the orthogonal energy conversion between heat and charge transfer in different mediums is free from such fundamental limitations. Moreover, the STE device is readily scalable by simply extending the area of a bilayer film, whereas scaling of the conventional TE module involves with series connections of alternating $p$–$n$ pitches[3]. Nonetheless, scaling of the STE device still necessitates the development of facile film processings of magnetic insulators.

Several inorganic magnetic insulators have been used for STE devices demonstrating effective STE energy conversion[4,8,9]. In particular, Yttrium iron garnet $Y_3Fe_5O_{12}$ (YIG) has been most widely used for STE applications because its low Gilbert damping constant ($\alpha = 2.3 \times 10^{-4}$) allows long-distance magnon

propagation[10]. However, inorganic magnetic insulator films are not appropriate for practical STE applications due to scaling problem, because they are difficult to grow into a large area and in need of high temperature processing for crystallization[3,11]. In contrast, organic or molecular films are generally grown at a lower temperature, and their flexible synthetic routes could offer scalable deposition techniques. Organic-based magnetic film, for example, vanadium tetracyanoethylene (V(TCNE)$_x$, $x \sim 2$) has shown effective spin-polarized carrier injection[12,13] as well as coherent magnon generation and spin pumping[14].

Prussian blue analog (PBA) is another family of molecule-based magnets with a molecular formula of AM$^1$[M$^2$(CN)$_6$]$_x\cdot n$H$_2$O (A is an alkali cation; M$^1$ and M$^2$ are transition metal ions). Substitution of transition metals at the M$^1$ and M$^2$ sites can produce a wide range of magnetic transition temperatures. For example, Cr$^{II}$[Cr$^{III}$(CN)$_6$]$_x\cdot n$H$_2$O (Cr-PBA) showed $T_c$ up to 240 K[15–18] and V$^{II}$[Cr$^{III}$(CN)$_6$]$_x\cdot n$H$_2$O powder exhibited $T_c$ up to 376 K[19,20]. Thus, PBAs are viable alternative magnetic insulators in STE devices with the advantage of versatile synthesis amenable for large area deposition at room temperature. Moreover, the molecule-based magnet may present weak spin–orbit coupling and lack of spin–lattice scatterings in addition to low thermal conductivity, which could lead to effective propagation of thermally excited magnons while retaining a greater temperature gradient.

In this work, we introduce a molecule-based magnet, Cr-PBA, as an alternative magnetic insulator for the magnon-mediated thermal-to-electrical energy conversion. The growth of Cr-PBA was done at room temperature by employing the electrochemical deposition (ECD) method, which could offer scalable production of thin films. A Cr (10 nm) thin film was used as a working electrode for ECD of Cr-PBA. The developed Cr-PBA/Cr bilayer can be directly utilized as an STE device with a seamless interface for effective spin pumping. The characterization of the developed

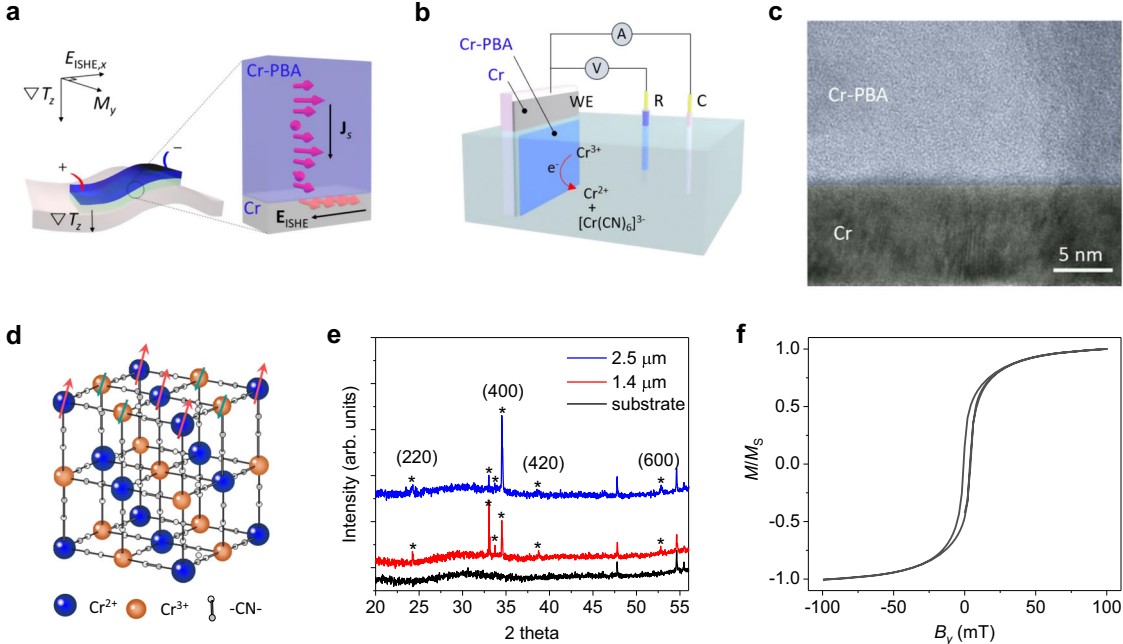

**Fig. 1 STE coating and characteristics of the Cr-PBA thin films. a** Schematic illustrations of STE energy conversion upon the vertical temperature gradient ($\nabla T_z$) and the mechanism of LSSE associated with thermally generated magnons and their conversion into a charge current via the ISHE. **b** Sketch of the ECD set-up for the Cr-PBA coating using a Cr thin film (10 nm) as a working electrode (WE), Pt counter electrode (C), and Ag/AgCl reference cell (R). **c** The cross-sectional TEM image of a Cr-PBA/Cr bilayer deposited on an oxidized silicon substrate. **d** Crystal structure of Cr[Cr(CN)$_6$] displaying a ferrimagnetic configuration (antiparalleled Cr$^{2+}$ ($S = 2$, $t_{2g}^3e_g^1$) and Cr$^{3+}$ ($S = 3/2$, $t_{2g}^3$) spins via the superexchange coupling). **e** XRD spectra of Cr-PBA films with deposition time of 600 and 1200 s, which are 1.4 and 2.5 µm thick, respectively. **f** The magnetic hysteresis curve of the developed Cr-PBA film measured for in-plane applied magnetic field at 100 K.

STE device was done based on the configuration of the longitudinal spin SSE (LSSE)[21–23], where the applied vertical temperature gradient is converted into the longitudinal electric voltage $V_{LSSE}$. We show that our Cr-PBA/Cr bilayer, which has 5 mm length, 100 µm width, and 1.4 µm thickness of Cr-PBA, produces a robust spin Seebeck voltage $\Delta V_{LSSE}/\Delta T \sim 64.9 \pm 3.13$ µV/K at 100 K. We also confirmed the generation, transport, and detection of magnons in the Cr-PBA/Cr bilayer via ferromagnetic resonance (FMR) and FMR-driven ISHE experiments. The Cr-PBA films were found to have an extremely low Gilbert damping constant. Our study shows excitations and transfers of magnons in this hybrid magnet are very efficient, suggesting molecule-based magnets, along with their synthetic versatility, could be outstanding alternatives for various applications of spin caloritronics as well as magnon spintronics.

## Results

### Fabrication and characterization of Cr-PBA/Cr bilayers.
Figure 1a shows the schematic illustration of an STE coating over a wide area and the electrical energy harvesting from the thermally excited magnons. Here, the vertical temperature gradient ($\nabla T_z$) generates vertical propagation of magnons in a magnetic insulator, which induces thermally pumped spin angular moments at the interface. Then, the transferred vertical spin flow will be converted into a longitudinal charge current via ISHE, $\mathbf{J}_{LSSE} \propto \mathbf{J_s} \times \mathbf{\sigma}$ ($\mathbf{J_s}$ and $\mathbf{\sigma}$ denote a thermally generated spin current vector and a spin polarization vector of electron, respectively). Figure 1b displays a schematic experimental set-up for ECD of the Cr-PBA film on a Cr metal. We employed a Cr thin film (10 nm) as a working electrode because it could provide relatively high spin–charge conversion, as evidenced in various literatures[24,25]. Heavy metals, such as Pt and Pd, are not appropriate for the working electrode due to a catalytic effect during the deposition. The Cr-PBA films are deposited through the reaction between [Cr(CN)$_6$]$^{3-}$ and labile Cr$^{2+}$ in the aqueous solution of K$_3$Cr(CN)$_6$ and CrCl$_3$·H$_2$O. A cross-sectional transmission electron microscope image of the developed Cr-PBA/Cr heterojunction is shown in Fig. 1c displaying a sharp interface. Further details of material preparation and characterization are described in "Methods" and Supplementary Note 1. Cyclic voltammetry curves for the observation of reductive reaction are shown in Supplementary Fig. 1. Cr-PBA has a structure consisting of interpenetrating face-centered cubic sublattices with ferrimagnetic spin alignments between Cr$^{2+}$ ($S=2$) and Cr$^{3+}$ ($S=3/2$) (Fig. 1d). X-ray diffraction spectra of Cr-PBA films are displayed in Fig. 1e. Results confirm a face-centered cubic phase of Cr-PBA with main peaks 24.2° (220), 34.5° (400), 38.7° (420), and 52.8° (600)[26]. The surface morphology of a Cr-PBA film probed by atomic force microscopy (AFM) is displayed in Supplementary Fig. 2. The magnetic hysteresis of the developed Cr-PBA film exhibits a coercivity of ~0.25 mT for in-plane applied magnetic field at 100 K (Fig. 1f). The Curie temperature of the developed Cr-PBA determined by Arrott plot method was $T_c \sim 230$ K (see Supplementary Fig. 3). Supplementary Fig. 4 displays a magnetization versus $T^{3/2}$ plot exhibiting a good linearity following the Bloch $T^{3/2}$ law. Fitting with $M(T) = M_0(1 - aT^{3/2})$ provides the value of slope $a \sim 2.6 \times 10^{-4}$ K$^{-3/2}$. This value of the slope is much higher than those observed in Ni ($7.5 \times 10^{-6}$ K$^{-3/2}$)[27] and YIG ($5.8 \times 10^{-5}$ K$^{-3/2}$)[28]. Thus, the excitations of magnons with low wave vectors would be more effective in our Cr-PBA film.

### STE characterization of Cr-PBA/Cr hybrid heterostructures.
For the characterization of spin thermoelectricity of the developed heterojunction devices, a heat gradient was applied by Joule heating from an Au line on top of the device as shown in Fig. 2a.

The on-chip Au line was also simultaneously used as a temperature sensor[11,23]. An electrical insulation between the top Au heater and Cr-PBA/Cr bilayer was done by the insertion of Al$_2$O$_3$ (130 nm)/Parylene (400 nm) films. The bottom of a device substrate (Si) was in thermal contact with a heat reservoir. Figure 2b shows the measured LSSE signals ($V_{LSSE}$) as a function of $zy$ angle of the applied magnetic field (1 T). $V_{LSSE}$ exhibits a sinusoidal behavior with the maximum magnitude when $\mathbf{B}$ and $\nabla T_z$ are perpendicular to each other because it is proportional to $|\mathbf{J_s} \times \mathbf{\sigma}|$. Figure 2c shows $V_{LSSE}$ as a function of applied $B_y$ measured for different $I_{heater}$. $V_{LSSE}$ exhibits a small coercivity and nearly saturates at ~300 mT. With increasing $I_{heater}$, $V_{LSSE}$ also increases. The obtained $\Delta V_{LSSE}$ ($\Delta V_{LSSE} = |V(0.5\ T) - V(-0.5\ T)|$) for $I_{heater} = 20$ mA was 47.3 µV.

The spin Seebeck coefficient ($S_{LSSE}$) of our STE devices is defined as $S_{LSSE} = E_{ISHE}/\nabla T = (\Delta V_{LSSE}/L)/(\Delta T/d)$, where is $L$ the distance between voltage probes and $d$ is the thickness of the Cr-PBA film. The estimation of the applied temperature gradient ($\nabla T$) in each layer of our STE device was done by employing the Fourier's law ($q_x = -\kappa \frac{dT}{dx}$) of a heat conduction. Here, we ignored the interfacial temperature drop as the interfaces are atomically contacted. The estimation of the temperature gradient on Cr-PBA may include slight uncertainty as the relative thickness of Cr-PBA is very thin compared to the interval where we measured temperature[29]. The unknown thermal conductivity ($\kappa$) of the Cr-PBA film was characterized by employing a differential $3\omega$ method[30,31] (Supplementary Note 2 and Supplementary Fig. 5). The estimated $\kappa$ of Cr-PBA is ~2.17 ± 0.01 W/mK at 100 K, which is much lower than those of inorganic magnets. Thus, the studied molecule-based magnetic film will be effective in keeping the applied temperature gradient. Details of temperature calibration are described in Supplementary Note 2, Supplementary Figs. 6 and 7, and Supplementary Table 1. The estimated $\Delta T$ on the Cr-PBA film and measured $\Delta V_{LSSE}$ are all directly proportional to Joule heating power $\sim I^2$ (Supplementary Fig. 8). Thus, $\Delta V_{LSSE}$ as a function of $\Delta T$ in Cr-PBA exhibits excellent linear dependence (Fig. 2d). The obtained $\Delta V_{LSSE}/\Delta T$ is ~64.9 ± 3.13 µV/K at 100 K. Then, the $S_{LSSE}$ of our device was calculated to be 18.2 nV/K. Supplementary Fig. 9 shows the estimated $S_{LSSE}$ for the devices with several different thicknesses of Cr-PBA. Results display a general tendency of the enhancement of $S_{LSSE}$ with increasing thickness of Cr-PBA. In order to confirm the origin of the observed $\Delta V_{LSSE}$, anomalous Hall effect was also measured. Results clearly exclude the possibility of a proximity-induced ferromagnetism in the Cr layer and its contribution to $\Delta V_{LSSE}$ (Supplementary Fig. 10).

### Temperature and field dependence of STE characteristics.
Figure 3a displays $V_{LSSE}$ ($B_y$) measured at various temperatures. The $\Delta V_{LSSE}$, defined as $|V_{LSSE}(+B_y) - V_{LSSE}(-B_y)|$ at $B_y = 0.5$ T, as a function of temperature is plotted in Fig. 3b. As temperature increases, the $\Delta V_{LSSE}$ initially increases gradually and then it starts to decrease after exhibiting a peak at around 60 K. In the LSSE configuration, the temperature dependence of $\Delta V_{LSSE}$ relies on the magnon excitation and its vertical propagation in a magnetic thin film[32,33]. As temperature decreases, the number of excited magnons gradually decrease following Bloch $T^{3/2}$ law but the magnon characteristic length ($\xi$) significantly increases. Within the atomic spin model, only the magnons excited at distances smaller than $\xi$ contribute to $\Delta V_{LSSE}$[34]. Thus, the increase of $\xi$ with lowering $T$ leads to the enhancement of $\Delta V_{LSSE}$ because more number of magnons involve in spin pumping at lower temperature. In our Cr-PBA film, $\xi$ becomes comparable to the thickness of the film below 60 K. This size effect together with Block $T^{3/2}$ law lead to the decrease of $\Delta V_{LSSE}$ below 60 K. We note

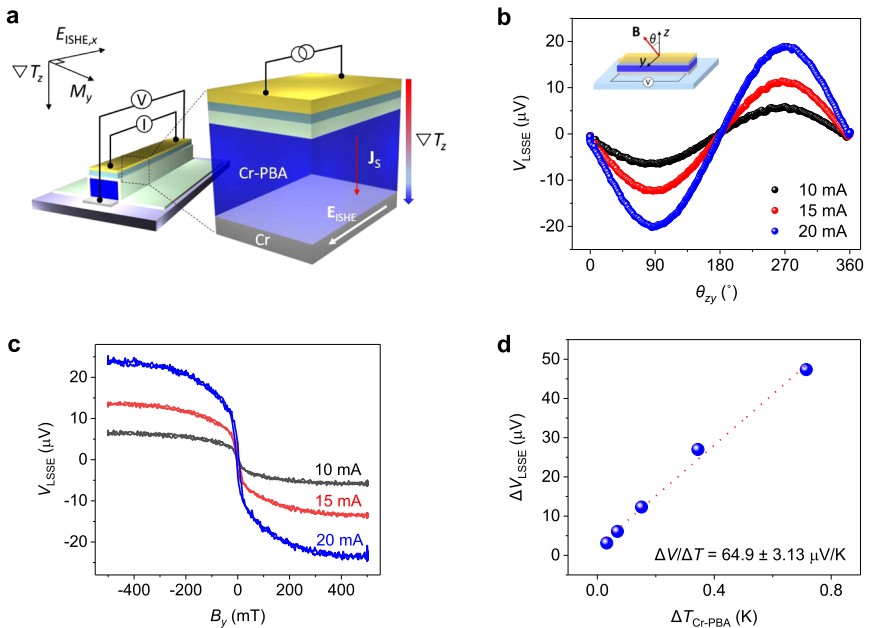

**Fig. 2 LSSE characterization for Cr-PBA/Cr heterojunction devices. a** A schematic illustration for the LSSE characterization of the Cr-PBA/Cr STE device. The temperature gradient was applied by Joule heating of a top Au line. The vertical flow of magnons pumps a pure spin current to the adjacent Cr layer. Then, the induced spin flow is converted into a longitudinal charge current producing electric field of $E_{ISHE}$. **b** $V_{LSSE}$ as a function of $zy$ angle of the applied magnetic field (1 T) measured with different heating currents $I_{heater}$ = 10, 15, and 20 mA at 100 K. Measurements were done for the Cr-PBA (1.4 μm)/Cr (10 nm) STE device. **c** $V_{LSSE}$ upon sweeping the applied magnetic field measured with different heating currents $I_{heater}$ = 10, 15, and 20 mA at 100 K. **d** $\Delta V_{LSSE}$ as a function of the estimated $\Delta T$ in a Cr-PBA film displaying a linear behavior. The obtained value of the slope is $\Delta V_{LSSE}/\Delta T \sim 64.9 \pm 3.13$ μV/K at 100 K.

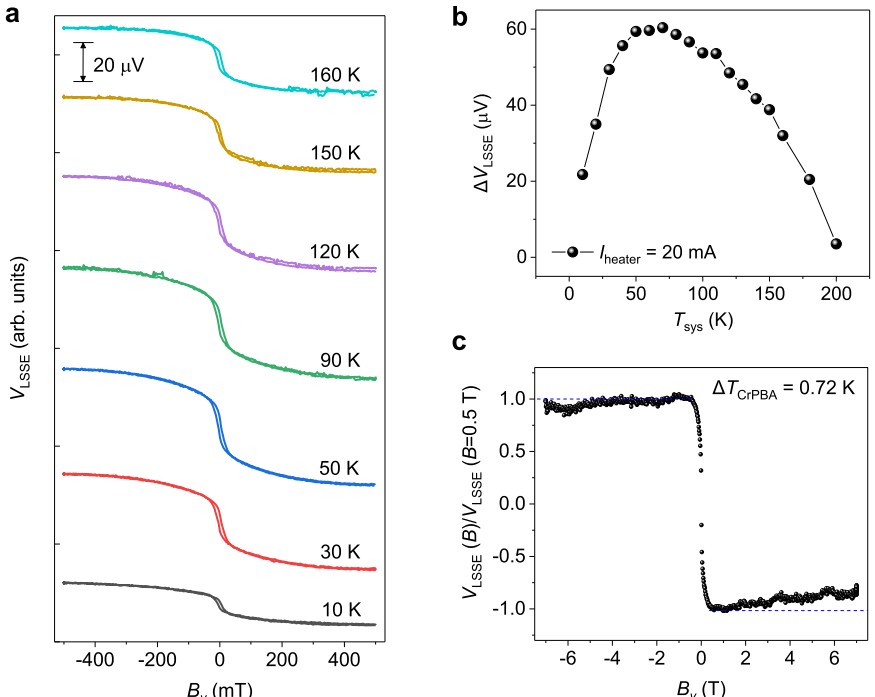

**Fig. 3 Temperature- and field dependence of $V_{LSSE}$. a** Temperature dependence of $V_{LSSE}$ measured with a Joule heating current, $I_{heater}$ = 20 mA ($\Delta T_{Cr\text{-}PBA}$ = 0.72 K). Measurements were done for the Cr-PBA (1.4 μm)/Cr (10 nm) STE device with a sweeping magnetic field ($B_y$) between −0.5 and 0.5 T. $V_{LSSE}$ ($B_y$) curves are vertically shifted for clarity. **b** Temperature dependence of $\Delta V_{LSSE} = |V_{LSSE}(+B_y) - V_{LSSE}(-B_y)|$ ($B_y$ = 0.5 T), displaying a peak at around 60 K. **c** High magnetic field dependence of $V_{LSSE}$ measured at 100 K with $I_{heater}$ = 20 mA ($\Delta T_{Cr\text{-}PBA}$ = 0.72 K).

that the peak of temperature-dependent $V_{LSSE}$ shifts to lower temperature with increasing the thickness of Cr-PBA (Supplementary Fig. 11a). This behavior is consistent with the size effect associated with magnon propagation length. We also measured temperature-dependent $\kappa$ of Cr-PBA to observe possible impacts of phonon on $V_{LSSE}(T)$. Results show that the $\kappa$ (T) of Cr-PBA increases with increasing temperature within the measurement window (up to 300 K) (Supplementary Fig. 11b).

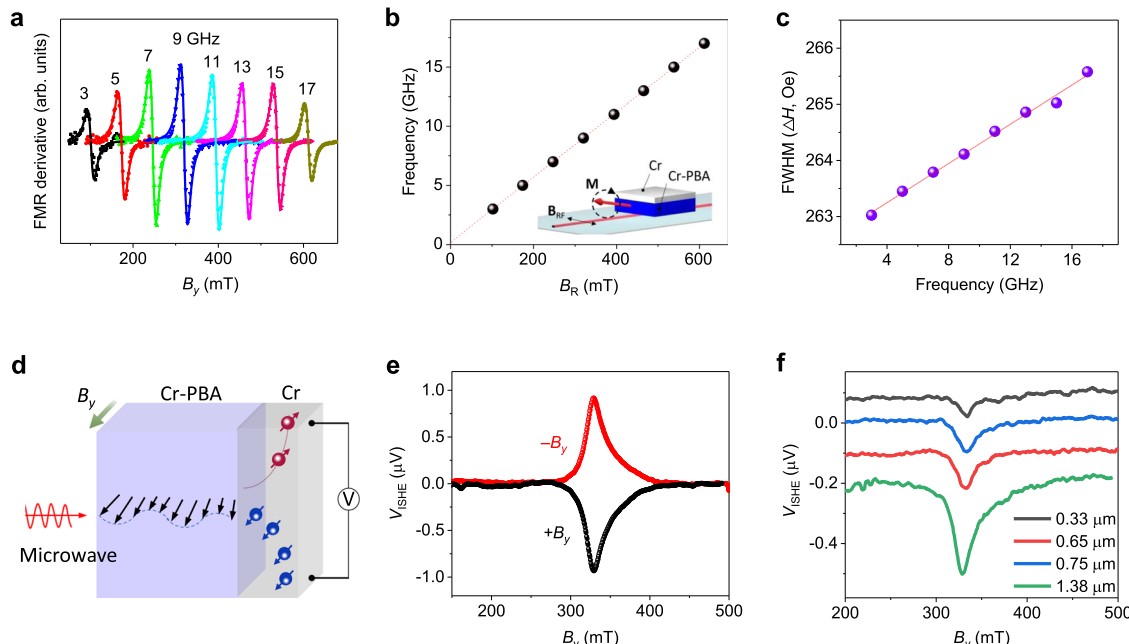

**Fig. 4 FMR and FMR-driven ISHE results for Cr-PBA/Cr heterojunctions. a** First derivative FMR spectra of the Cr-PBA (1.4 μm)/Cr (10 nm) heterojunction at various microwave frequencies measured at 100 K. The recorded data are fitted by using the derivative of the Lorentzian function. **b** Frequency dependence of resonance field $B_R$. Fitting with the Kittel equation produces $M_s = 12.676$ kA/m and $g$-factor = 1.96. **c** FWHM as increasing the frequency of r.f. field. The slope of a linear fit produces a low effective Gilbert damping constant, $\alpha_{eff} \sim 7.5 \times 10^{-4}$. **d** A schematic illustration of the spin pumping from microwave driven magnon propagation in the Cr-PBA film and its conversion into an electrical current in the Cr layer via ISHE. **e** $V_{ISHE}(B)$ measured for the Cr-PBA (1.4 μm)/Cr (10 nm) bilayer upon applying 9 GHz of microwave. **f** $V_{ISHE}(B)$ of Cr-PBA/Cr heterojunctions with various Cr-PBA thicknesses.

Figure 3c displays the magnetic field dependence of $\Delta V_{LSSE}$ with applying magnetic fields up to 7 T. We note that the Nernst effect in the Cr layer produces field-dependent voltage ($V_x(B_y)$) having the same symmetry of $V_{LSSE}$ (see Supplementary Fig. S12a–c). Thus, we subtracted the Nernst effect from the measured $V_x(B_y)$ assuming that the Nernst effect in the Cr layer is temperature independent. As the magnitude of the magnetic field increases from 0.5 to 7 T, the LSSE signal is monotonically suppressed. The ratio of $\Delta V_{LSSE}$ change at 7 T, $[(V_{LSSE}$ (7 T) − $V_{LSSE}$ (0.5 T))/$V_{LSSE}$ (0.5 T)] × 100%, is calculated to be ~14%, which is similar to that observed in YIG[22,28,35]. Previous studies with YIG attributed the high-field suppression of $V_{LSSE}$ to the suppression of sub-thermal magnon[28,36] (Supplementary Fig. 12d). However, the high-field suppression of the estimated $V_{LSSE}$ in our studies could be due to the variation in the Nernst effect because the applied $\Delta T$ could be altered at different temperature (Supplementary Note 2). The field-dependent $V_{LSSE}$ was also measured at different temperatures as shown in Supplementary Fig. 13. Results display no consistent tendency of the field-dependent suppression as varying system temperature. We note that the changes of resistance in the Au heater line and the Cr layer in a high magnetic field are negligible and not associated with high-field suppression of $V_{LSSE}$ (Supplementary Fig. 14).

**Spin dynamics and pumping at the Cr-PBA/Cr heterostructures**. The generation, propagation, and detection of magnons in the Cr-PBA/Cr heterostructure were further studied through FMR and FMR with ISHE (FMR-ISHE) experiments. Figure 4a shows FMR spectra of a Cr-PBA (1.4 μm)/Cr (10 nm) heterojunction measured at various microwave frequencies. The frequency dependence of a resonance field ($B_R$) shown in Fig. 4b

well follows the Kittel equation, $\nu = \frac{\gamma}{2\pi} \left[ B_R \left( B_R + \mu_0 M_s \right) \right]^{1/2}$, where $\nu$ is microwave frequency, $\gamma$ is the gyromagnetic ratio, and $M_s$ is a saturation magnetization. The frequency dependence of full-width at half-maximum (FWHM) shown in Fig. 4c gives the estimation of the effective Gilbert damping constant ($\alpha_{eff}$) following the relation, $\Delta B = \Delta B_0 + \frac{4\pi\alpha_{eff}\nu}{\sqrt{3}\gamma}$. Here, $\Delta B_0$ denotes an inhomogeneous broadening by structural imperfections. Since the Cr-PBA film is grown on the Cr metal layer which absorbs spin angular momentum at the interface, the effective Gilbert damping in Cr-PBA/Cr becomes larger than the intrinsic damping in Cr-PBA. The obtained $\alpha_{eff}$ is ~7.5 × 10$^{-4}$ for the Cr-PBA (1.4 μm)/Cr (10 nm) heterojunction. The intrinsic damping constant ($\alpha_0$) of the Cr-PBA film and the spin mixing conductance ($g_{eff}^{\uparrow\downarrow}$) of the heterojunction can be estimated from the relation[37], $\alpha_{eff} = \alpha_0 + \Delta\alpha = \alpha_0 + \frac{g\mu_B}{4\pi M_s d} g_{eff}^{\uparrow\downarrow}$, where $\Delta\alpha$ is the additional Gilbert damping caused by spin pumping at the interface and $d$ is the thickness of Cr-PBA. The thickness dependence of $\alpha_{eff}$ is displayed in Supplementary Fig. 15. The obtained $\alpha_0$ is $(2.4 \pm 0.67) \times 10^{-4}$ and $g_{eff}^{\uparrow\downarrow}$ is $(6.5 \pm 0.52) \times 10^{18}$ m$^{-2}$, which are comparable to those of the epitaxial YIG and YIG/Pt[10,37]. The observed small damping indicates that the generated magnons in the Cr-PBA film will be effectively delivered to the interface with low loss, relevant for effective spin thermoelectricity as well as magnon spintronics.

In order to investigate the conversion of spin angular momentums into an electric current at the Cr-PBA/Cr interface further, we measured FMR-driven ISHE as illustrated in Fig. 4d. The transferred spin angular momentums at the interface induce a spin current in the adjacent Cr layer. Then, the conversion from spin to charge flow via the ISHE allows electrical detection of microwave pumped magnons. Because $\mathbf{E}_{ISHE} \propto \mathbf{J}_s \times \boldsymbol{\sigma}$, a

maximum magnitude of $V_{ISHE}$ can be obtained when magnetic field $\mathbf{B}$ is normal to the direction of voltage probes. Figure 4e displays measured $V_{ISHE}$ by applying continuous microwave $f = 9$ GHz to the Cr-PBA (1.4 μm)/Cr (10 nm) heterojunction. The FWHM and $B_R$ of $V_{ISHE}$ are consistent with those of FMR spectra shown in Fig. 4a. $V_{ISHE}$ reverses polarity when reversing the field direction (Fig. 4e). These behaviors clearly suggest that the observed $V_{ISHE}$ originates from the FMR-generated magnon flow in the Cr-PBA film. As increasing $d$, the FMR-driven $V_{ISHE}$ substantially increases as shown in Fig. 4f. Using the obtained $g_{eff}^{\uparrow\downarrow}$, we can estimate the transferred spin current density $j_s^0$ at the interface (Supplementary Note 3). Then, the spin Hall angle ($\theta_{SHE}^{Cr}$) of the Cr layer can be obtained from the relation, $V_{ISHE} = \frac{2e}{\hbar}\frac{\theta_{SHE}^{Cr}\lambda_s^{Cr}\omega j_s^0}{\sigma^{Cr}d^{Cr}}\tanh\left(\frac{2d^{Cr}}{\lambda_s^{Cr}}\right)$, where $\lambda_s^{Cr}$ and $\sigma^{cr}$ are the spin diffusion length and conductivity of the Cr layer, respectively. Taking $\lambda_s^{Cr} = 2.1$ nm from the literature[25], we obtain the spin Hall angle ($\theta_{SHE}^{Cr}$) ~ −0.014. This value is within the range of $\theta_{SHE}^{Cr}$ values in the literatures[24,25]. Here, improving spin–charge conversion with other compatible electrodes, such as transition metals[38,39], alloy metals[40,41], and topological insulators[42], would significantly enhance the spin Seebeck coefficient of Cr-PBA. In short, our FMR-ISHE studies confirm an effective generation of a magnonic spin flow in Cr-PBA and spin-pumping process at the Cr-PBA/Cr interface.

## Discussion

In conclusion, we introduced a new class of magnetic materials for spin caloritronics. The studied molecular magnetic film has several advantageous characteristics over inorganic magnetic insulators in terms of STE applications. Conventional electro-deposition at room temperature was successfully employed for the fabrication of the Cr-PBA-based STE device. This deposition technique can be easily adopted for the large area and mass production of thin film, which can boast an important merit of STE, that is, large-area scalability. Various other methodologies, such as painting and printing, can be also utilized for developing the PBA film. The generation and transfer of magnons are essential processes for STE energy harvesting as well as magnon information technology. Excitations of low-energy magnons in this class of magnet are much stronger than those in the typical inorganic magnets. The obtained low Gilbert damping constant, comparable to that of the epitaxial YIG film, grants transfer of thermally excited magnons over the long distance with low loss. The determined low thermal conductivity in the studied molecule-based magnetic film is an accessory benefit for STE energy harvesting because it assists in maintaining a higher temperature gradient across the film. In short, our study showed that molecule-based magnetic films could be outstanding alternatives for STE energy harvesting as well as promising platforms for the generation and transmission of magnons in various spin-based electronic applications.

## Methods

**Film deposition and characterization**. For the fabrication of Cr-PBA/Cr heterojunctions, a Cr metal thin film (10 nm) for a working electrode was first deposited on a SiO$_2$ (300 nm)/$p$-Si (500 μm) substrate by e-beam evaporation under the base pressure of ~$7 \times 10^{-7}$ Torr. The electrochemical processes were carried out using a potentiostat with a reference electrode (Ag/AgCl) and a Pt counter electrode. For a Cr-PBA deposition, the aqueous solution of 7.5 mM CrCl$_3$·H$_2$O and 5 mM K$_3$[Cr(CN)$_6$] was used. The coating of a Cr-PBA film was obtained by electrochemical reduction (at a fixed potential $E = -0.88$ V vs. Ag/AgCl reference electrode) of Cr$^{3+}$ in an aqueous solution containing [Cr(CN)$_6$]$^{3-}$ anions. The Cr$^{2+}$ species formed at the surface of the Cr metal electrode react with the [Cr(CN)$_6$]$^{3-}$ developing an insoluble PBA coating. The deposition rate of the PBA film in our processing was typically ~140 nm/min. For the study of STE characterizations, the electrodepositions of PBA films were done for 10 minutes

producing a thickness of 1.4 μm. The surface roughness of the Cr-PBA film was measured using atomic force microscopy (DI-3100, Veeco) and the cross-sectional image of the bilayer was investigated by normal-TEM (ZEM-2100). The crystallization of the film was probed by using a X-ray diffraction (D8 ADVANCE, Bruker AXS). The magnetic properties of the Cr-PBA film were measured by using a superconducting quantum interference device-vibrating sample magnetometer (SQUID-VSM, Quantum Design).

**Spin-TE device fabrication with an on-chip heater**. We used a heterojunction of Au (20 nm)/Al$_2$O$_3$ (130 nm)/Parylene (400 nm)/Cr-PBA (1.4 μm)/Cr (10 nm) for the characterization of LSSE in the Cr-PBA/Cr STE device. After the deposition of Cr-PBA on the Cr film, 400 nm of parylene was deposited to protect the entire sample by using a standard parylene coater (Alpha plus). For the fabrication of the studied device, the reactive ion etching was done with O$_2$ and Cl$_2$ gas by using a thick photoresist (AZ9660) as a protective buffer patterned for a dimension of 5 mm length and 100 μm width. After removing the buffer photoresist by acetone, the additional insulating layer of Al$_2$O$_3$ (130 nm) and the top Au heater (20 nm) patterned into the same dimension of Cr-PBA (5 mm length and 100 μm width) were successively deposited by e-beam evaporation.

**STE characterization with temperature calibration**. The current source for the Joule heating was applied by using a Keithley 2636 sourcemeter. The temperature of the Au layer was estimated based on the temperature dependence of the electrical resistivity of the Au layer (Supplementary Fig. 6a). Temperature stabilization of the top Au layer depends on the heating current and usually takes less than a few minutes (Supplementary Fig. 6b). LSSE measurements were performed after the temperature of the top Au layer was sufficiently stabilized. For the measurement of the thermal conductivity of the Cr-PBA film, we used the differential 3ω method by using Keithley 6221 ac sourcemeter and SR830 Lock-in amplifier to measure harmonic ac voltage. Measurements of the thermal conductivity are further described in Supplementary Note 2. Details of temperature calibration and SSE characterization are also explained in Supplementary Note 2. Under the applied temperature gradient, the induced voltage from SSE and ISHE was measured by using a Keithley 2182 nanovoltmeter. All measurements for LSSE were performed in a Physical Property Measurement System (PPMS) (Quantum Design) with varying temperature and magnetic field.

**FMR and FMR-ISHE measurements**. For the FMR measurements, we used the broadband FMR of the PPMS option with a coplanar waveguide and a NanOsc Phase FMR spectrometer operated in the frequency range 2–19 GHz. For the FMR-driven ISHE measurements, we used coplanar waveguide FMR with contact pads for the detection of ISHE (KBSI and RNDWARE Co. Ltd). Devices used for FMR-ISHE studies have a channel size of 1 mm width and 2 mm length with gold contact pads at both sides. Voltages generated by FMR-ISHE were detected by using Keithley 2182 nanovoltmeter. All FMR and FMR-ISHE measurements were performed in the PPMS chamber.

## Data availability

The data that support the findings of this study are available from the corresponding author upon request.

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

## Acknowledgements

This work was supported by the National Research Foundation of Korea (NRF) grant funded by the Korea government (MSIT) (2017M3A7B4049172, 2017R1A2B4008286, and 2020M3F3A2A03082444). This work was also supported by the Ulsan National Institute of Science and Technology (No. 1.200095.01). J.P. and Y.J. acknowledge the support of the National Research Council of Science & Technology (NST) grant (No. CAP-16-01-KIST) by the Korean government (MSIP). J.S. acknowledges the support of the National Research Foundation of Korea (NRF) grant funded by the Korea government (MSIT) (2020R1C1C1011219).

## Author contributions

I.O. and J.-W.Y. conceived the project. I.O. performed materials growth and characterization, device fabrication, and electrical measurements. I.O, J.P. and Y.J. performed FMR and FMR-ISHE experiments. I.O. and J.S. performed 3ω measurements. I.O., J.P., and J.-W.Y. did data analysis. H.J., J.J., D.C., and M.-J.J. assisted materials growth, device fabrication, and characterization. B.-C.M. assisted data analysis on FMR and FMR-ISHE results. I.O. and J.-W.Y. wrote the manuscript. All authors discussed about the results and commented on the manuscript.

## Competing interests

The authors declare no competing interests.
