## [Peer Review File · Nature Communications]

Reviewers' Comments:

Reviewer #1:

Remarks to the Author:

In this paper, Oh et al. report the observation of the spin Seebeck effect (SSE) in a Cr-PBA/Cr hybrid structure, where Cr-PBA is a molecule-based magnetic material. Since most of conventional studies on spin caloritronics have been performed by using inorganic magnetic materials, such as garnet and spinel ferrites, this work offers a new direction in the study of SSE and related spintronic phenomena. This paper is worth publishing and the experimental results reported are systematic. However, the present version includes insufficient discussions, and major revisions are necessary. For further consideration for publication in Nature Commun., the following comments should be addressed.

(1) The authors used a Cr film not only for a working electrode for the electrochemical deposition but also for spin-charge current conversion. This idea is clever but also limits the choice of materials and thermoelectric performance of organic-magnet-based SSE devices. In fact, as described in the main text, typical materials with larger spin Hall angle, such as Pt and Pd, cannot be used for this method. The authors should list possible alternative electrodes that should have larger spin Hall angle.

(2) On the lines 88-89, the authors state "Results show that our Cr-PBA/Cr bilayer produces a significantly large spin Seebeck voltage $\Delta V_{\text{LSSE}}/\Delta T \sim 59.6 \text{ uV/K}$ at 60 K". However, $\Delta V_{\text{LSSE}}/\Delta T$ is dependent on the sample size and not suitable for characterizing SSE devices. The authors should describe the sample size along with the $\Delta V_{\text{LSSE}}/\Delta T$ value, or just replace it with S_{LSSE} including the size information.

(3) I am worried about the accuracy of the S_{LSSE} estimation. As shown in Supplementary Fig. 7, the authors neglect the interfacial thermal resistances between the layers. This assumption is reasonable when the total temperature difference is dominated by the temperature difference in a magnetic insulator layer, which is typically a substrate in SSE devices (see J. Phys.: Condens. Matter 26, 343202 (2014)). However, this assumption cannot be applied to the Cr-PBA/Cr hybrid structure because the Cr-PBA layer is thin and interfacial thermal resistance between organic and inorganic materials is very large in general.

(4) The authors estimated the temperature of the top Au layer based on the temperature dependence of the resistance of the Au layer. Why didn't the authors estimate the temperature of the bottom Cr layer in the same way? This measurement enables the direct estimation of the temperature difference between the top and bottom of the sample (Phys. Rev. B 95, 174401 (2017)). This is also useful to check the accuracy of the temperature profile determination based on the 3-omega method.

(5) The temperature dependence and high-field behavior of the SSE voltage depend not only on the magnon transport properties but also the sample thickness. The results and interpretation shown in this paper will be confirmed further if the authors can show the results for several samples with different thicknesses.

(6) As shown in Fig. 3c, the SSE voltage is suppressed linearly with increasing the magnetic field. This behavior is quite different from conventional experiments using YIG. To confirm that this field dependence of the SSE voltage originates from the suppression of sub-thermal magnons, the authors should show the high field behavior at different temperatures as well as the magnitude of the ordinary Nernst effect in the Cr layer.

(7) " $V_{\text{ISHE}} \propto J_s \times \sigma$ " on page 10 is mathematically incorrect, because the left-hand (right-hand) side is a scalar (vector) value.

(8) The fabrication of SSE devices based on the electrochemical deposition is very interesting. Is it possible to fabricate Cr-PBA/Cr multilayers? Such alternately-stacked multilayer films are known to exhibit the enhancement of SSE (Phys. Rev. B 92, 220407(R) (2015)).

(9) The units for magnetic fields and magnetization should be unified. The present version includes both the cgs Gauss and SI unit systems; the latter is better.

(10) On page 5 of Supplementary Information, why did the authors determine the Curie temperature of Cr-PBA based on the magnetization curve at 10 Oe. Since 10 Oe is smaller than the coercive force, the magnetization curve is affected by the hysteresis. The Curie temperature should be determined by the Arrott plot method.

(11) On the same page, the authors state "Thus, the excitations of magnons with low wave vectors are more effective in our Cr-PBA film. This behavior indicates strong suppression of sub-thermal magnons will occur at high magnetic field,...", which is based on the difference in the temperature dependence of magnetization between Cr-PBA and Ni. However, this discussion is too rough and I do not understand why the authors compare the magnetization curve of Cr-PBA with that of Ni. Due to the spectral-dependent nature of SSE, it is difficult to predict its behavior only from static magnetic properties.

(12) The measurement temperature for the 3-omega method should be described. The error bars should be added to Supplementary Fig. 5.

Reviewer #2:

Remarks to the Author:

The work by Oh et al. reports on the fabrication of vertical heterojunctions based on molecular Prussian blue analogue (PBA) compounds, their physical characterization and their possible use as a spin thermoelectric material.

The manuscript is clear, easy to read and the topic and results are of broad interest. I find it suitable for being published in Nature Communications once the following points are addressed:

1.- How many different samples were measured for each of the performed experiments? Is there any dependence of the measured properties with the PBA thickness?

2.- In the Supplementary Figure 4, the field cool magnetization is performed with an external magnetic field of 0.5 T (i.e., a considerable field). From the fitting of the data, the authors state "Fitting with $M(T) = 178 M_0(1 - aT^{3/2})$ provides the value of slope $a \sim 2.6 \times 10^{-4} \text{ K}^{-3/2}$. This value of slope is much higher than those observed in Ni ($7.5 \times 10^{-6} \text{ K}^{-3/2}$)³⁴ and YIG ($5.8 \times 10^{-5} \text{ K}^{-3/2}$)³². Thus, the excitations of magnons with low wave vectors are more effective in our Cr-PBA film, leading to strong suppression of VLSSE at high magnetic field." How does the magnetization look like with a smaller applied field (like 10 Oe or 100 Oe)? Is the same slope obtained?

Responses to the reviewer's comments

Reviewer #1

Reviewer's comments:

In this paper, Oh et al. report the observation of the spin Seebeck effect (SSE) in a Cr-PBA/Cr hybrid structure, where Cr-PBA is a molecule-based magnetic material. Since most of conventional studies on spin caloritronics have been performed by using inorganic magnetic materials, such as garnet and spinel ferrites, this work offers a new direction in the study of SSE and related spintronic phenomena. This paper is worth publishing and the experimental results reported are systematic. However, the present version includes insufficient discussions, and major revisions are necessary. For further consideration for publication in Nature Commun., the following comments should be addressed.

Our responses:

We greatly appreciate reviewer's a number of important comments and advices as well as positive remarks. Below is the detailed discussions and corrections we made in response to reviewer's comments.

#1. Reviewer's comments:

The authors used a Cr film not only for a working electrode for the electrochemical deposition but also for spin-charge current conversion. This idea is clever but also limits the choice of materials and thermoelectric performance of organic-magnet-based SSE devices. In fact, as described in the main text, typical materials with larger spin Hall angle, such as Pt and Pd, cannot be used for this method. The authors should list possible alternative electrodes that should have larger spin Hall angle.

Our responses:

We appreciate reviewer's comments for important considerations, which we should have listed.

- There are several candidates for the working electrode which have large spin Hall angle but are not expected to trigger hydrogenation reaction during the electrochemical deposition (ECD). Those could be **transition metals** (V, Ir, Gd, Ru [T. Wang et al. *Sci. Rep.* **7**, 1306 (2017); Y. Wu et al. *Adv. Mater.* **20**,1603031 (2017)]), **alloy metals** (W-Hf, Cu-Tb [K. Fritz et al. *Phys. Rev. B* **98**, 094433 (2018); Z. Xu et al. *ACS. Appl. Mater. Interfaces* **12**, 32898-32904 (2020)]) and **Topological Insulator** which has giant spin Hall angle [Z. Jiang et al. *Nat. commun.* **7**, 11458 (2016)]. Those conductors might act as working electrodes without any hydrogenation reaction at the reduction voltage.

In response to reviewer's comment, we added following phrase in the revised manuscript.

Line 235 in the revised manuscript:

“such as transition metals[T. Wang et al. *Sci. Rep.* 7, 1306 (2017); Y. Wu et al. *Adv. Mater.* 20,1603031 (2017)], alloy metals[K. Fritz et al. *Phys. Rev. B* 98, 094433 (2018); Z. Xu et al. *ACS. Appl. Mater. Interfaces* 12, 32898-32904 (2020)], and topological insulators[Z. Jiang et al. *Nat. commun.* 7, 11458 (2016)]”

#2. Reviewer's comments:

On the lines 88-89, the authors state "Results show that our Cr-PBA/Cr bilayer produces a significantly large spin Seebeck voltage $\Delta V_{\text{LSSE}}/\Delta T \sim 59.6 \mu\text{V}/\text{K}$ at 60 K". However, $\Delta V_{\text{LSSE}}/\Delta T$ is dependent on the sample size and not suitable for characterizing SSE devices. The authors should describe the sample size along with the $\Delta V_{\text{LSSE}}/\Delta T$ value, or just replace it with S_{LSSE} including the size information.

Our responses:

We appreciate reviewer's considerate comments. We agree with that we should have written the geometry of our Cr-PBA/Cr devices. In response to reviewer's comment, we added following phrase in the revised manuscript.

Line 89 in the revised manuscript:

“which has 5 mm length, 100 μm width, and 1.4 μm thickness of Cr-PBA”

#3. Reviewer's comments:

I am worried about the accuracy of the S_{LSSE} estimation. As shown in Supplementary Fig. 7, the authors neglect the interfacial thermal resistances between the layers. This assumption is reasonable when the total temperature difference is dominated by the temperature difference in a magnetic insulator layer, which is typically a substrate in SSE devices (see *J. Phys.: Condens. Matter* 26, 343202 (2014)). However, this assumption cannot be applied to the Cr-PBA/Cr hybrid structure because the Cr-PBA layer is thin and interfacial thermal resistance between organic and inorganic materials is very large in general.

Our responses:

We appreciate reviewer's considerate comments. As the reviewer pointed out, the temperature

drop at the interface could not be trivial. As reviewer mentioned and discussed in ref. (*J. Phys.: Condens. Matter* 26, 343202 (2014)), the interfacial temperature drop could be ignored if the most of temperature difference is applied in a magnetic insulator, which is not the case in our experiment. However, there is no proven experimental method to measure temperature discontinuity at the atomic interface. As long as the interface is atomically contacted without other impurities, the interfacial temperature drop could be negligible. According to the TEM image (Fig. 1c), Cr-PBA/Cr heterojunction forms sharp interface with no crack and no discontinuity. Thus, we assumed that the interfacial temperature drop could be ignorable when we estimated the temperature gradient applied on a magnetic insulator in our device. We would like to note that many pervious literatures even ignored the temperature drop at the contact made with the insertion of thermal grease [A. Kirihara et al. *Nat. Mat.* 17, 686-689 (2012); A. Sola et al. *Sci. Rep.* 9, 2047 (2019)].

In response to the reviewer's comments, we added following discussion in the revised manuscript.

Line 146 (page 7) in the revised manuscript:

“Here, we ignored the interfacial temperature drop as the interfaces are atomically contacted. The estimation of temperature gradient on Cr-PBA may include slight uncertainty as the relative thickness of Cr-PBA is very thin compared to the interval where we measured temperature [*J. Phys.: Condens. Matter* 26, 343202 (2014)].”

#4. Reviewer's comments:

The authors estimated the temperature of the top Au layer based on the temperature dependence of the resistance of the Au layer. Why didn't the authors estimate the temperature of the bottom Cr layer in the same way? This measurement enables the direct estimation of the temperature difference between the top and bottom of the sample (*Phys. Rev. B* 95, 174401 (2017)). This is also useful to check the accuracy of the temperature profile determination based on the 3-omega method.

Our responses:

We appreciate reviewer's considerate suggestion. As reviewer suggested, the Cr layer could also be used as a temperature sensor. However, there are a number of other effects coming in when we measure the resistance of Cr while heating the top Au layer. First, the spin Seebeck effect produce LSSE signal, which affects the measurement of the resistance of Cr. In addition, we should also apply source current in Cr layer to measure resistance, which could create spin Peltier effect. The interplay between spin Seebeck effect and spin Peltier effect make the precise measurement of the resistance of Cr layer difficult. In general, metal films, such as Au and Pt are good yardsticks for the temperature calibration as their film have good linearity in their resistivity over the wide range of temperature. But thin films of Cr are typically poly-

crystalline and generally exhibit unconventional temperature dependence due to various interaction effects of electrons in weakly disordered system (see M. Ohashi et al. *Phys. Lett. A* **380**, 3133 (2016)). Thus, we estimated the temperature gradient on Cr-PBA from the total temperature difference between the top Au layer and the bottom heat reservoir.

#5. Reviewer's comments:

The temperature dependence and high-field behavior of the SSE voltage depend not only on the magnon transport properties but also the sample thickness. The results and interpretation shown in this paper will be confirmed further if the authors can show the results for several samples with different thicknesses.

Our responses:

We appreciate reviewer's valuable comment for adding depth to our paper. As reviewer's suggested, we performed the measurements for the temperature dependence and high-field behavior of the SSE voltage with three different thicknesses of Cr-PBA.

- Temperature dependence of LSSE voltage with different thickness

We observed that the peak of temperature dependent V_{LSSE} shifted to higher temperature with decreasing thickness of Cr-PBA and almost suppressed for 0.5 μm thick film. This behavior is similar to what have been observed in YIG film [E. Guo et al., *Phys. Rev. X* **6**, 031012 (2016)]. As we discussed in the manuscript, the peak of temperature dependent V_{LSSE} occurs when the thickness of Cr-PBA becomes comparable to the magnon propagation length, which increases with decreasing temperature. We also measured temperature dependence of the thermal conductivity of Cr-PBA to observe possible impact of phonon on $V_{LSSE}(T)$. The thermal conductivity of Cr-PBA film increases as increasing temperature within the measurement window (up to room temperature) (see Supplementary Fig. S11b). Thus, we can confirm that the observed peak of $V_{LSSE}(T)$ is a size effect associated with magnon propagation length.

- Field dependence of LSSE with different thickness

As reviewer suggested, we also measured high-field behavior of V_{LSSE} up to 7 T for three different thicknesses. The high-field suppression of V_{LSSE} was initially attributed to strong suppression of sub-thermal magnons. However, as reviewer suggested in the next comments, we checked the transverse field-dependent voltage ($V_x(B_y)$) while applying vertical temperature gradient (∇T_z) even above the magnetic transition temperature of Cr-PBA. We noticed that Nernst effect produces field-dependent voltage ($V_x(B_y)$), which has the same symmetry of V_{LSSE} . The observed Nernst effect was displayed in Fig. S12a. Thus, we subtracted the Nernst effect from the high magnetic field dependent V_{LSSE} signal. The obtained high-field suppression ratio of the LSSE signal increases as increasing thickness of the Cr-PBA film, which is consistent with previous work [U. Ritzmann, *Phys. Rev. B*, **92**, 174411 (2015), T. Kikkawa, *Phys. Rev. B*, **92**, 064413 (2015)]. We also monitored field-dependent resistance of Au heater line and Cr

metal line because magnetoresistance of them could affect the measured LSSE signal. The field-induced resistance changes of Cr and Au were only 0.043, 0.037 %, respectively, indicating a negligible impact on the observed high-field suppression of the LSSE signals.

In response to reviewer's comments, we added further discussions on the temperature- and field-dependence of LSSE on main text (in page 9-10) and Supplementary information (in page 16-17). We also added Supplementary Fig. 11 and 12.

#6. Reviewer's comments:

As shown in Fig. 3c, the SSE voltage is suppressed linearly with increasing the magnetic field. This behavior is quite different from conventional experiments using YIG. To confirm that this field dependence of the SSE voltage originates from the suppression of sub-thermal magnons, the authors should show the high field behavior at different temperatures as well as the magnitude of the ordinary Nernst effect in the Cr layer.

Our responses:

We appreciate reviewer's critical comment.

We greatly appreciate reviewer's valuable comments and suggestion. Indeed, the linear suppression of LSSE signal with increasing magnetic field was also observed for YIG in various literatures [E. Guo et al. *Phys. Rev. X* **6**, 031012 (2016); U. Ritzmann et al. *Phys. Rev. B* **92**, 174411 (2015); T. Kikkawa et al. *Phys. Rev. B* **92**, 064413 (2015); H. Jin et al. *Phys. Rev. B* **92**, 054436 (2015)], where they attributed to the suppression of low-energy magnon. Such linear suppression of LSSE signal was observed to be much stronger in our device with Cr-PBA. Thus, we attributed such behavior to the strong suppression of low-energy magnon at high magnetic field. Following the reviewer's advice, we checked the Nernst effect at temperature higher than magnetic critical temperature of Cr-PBA. Results show that the Nernst effect produces field-dependent voltage ($V_x(B_y)$) having the same symmetry of V_{LSSE} . Thus, we subtract the Nernst effect from the field-dependent LSSE signals. We measured the field dependent LSSE signal at different temperatures as shown in Supplementary Fig. 13. Results exhibit no consistent tendency of the field-dependent suppression ratio as varying system temperature.

In response to reviewer's comments, we added Supplementary Fig. 13 and following sentences in the revised manuscript.

Line 191 (page 10) in the revised manuscript:

“The field dependent V_{LSSE} was also measured at different temperatures as shown in Supplementary Fig. 13. Results display no consistent tendency of the field-dependent suppression as varying system temperature.”

#7. Reviewer's comments:

" $V_{\text{ISHE}} \propto J_s \times \sigma$ " on page 10 is mathematically incorrect, because the left-hand (right-hand) side is a scalar (vector) value.

Our responses:

We greatly appreciate reviewer's careful proof reading. We changed the V_{ISHE} into \mathbf{E}_{ISHE} and also changed every vector value as a boldface form throughout the manuscript.

#8. Reviewer's comments:

The fabrication of SSE devices based on the electrochemical deposition is very interesting. Is it possible to fabricate Cr-PBA/Cr multilayers? Such alternately-stacked multilayer films are known to exhibit the enhancement of SSE (Phys. Rev. B 92, 220407(R) (2015)).

Our responses:

We appreciate reviewer's suggestion. As reviewer commented, adding stacked multilayer could enhance the V_{LSSE} . For the fabrication of our devices, we deposited Cr layer in a vacuum chamber, while Cr-PBA was grown in solution. Thus, stacking multilayer involves in switching deposition environment back and forth. Such process is not desirable for the fabrication of clean interface, which is critical for the effective conversion between spin flow and charge flow. In order to obtain high-quality multilayer, further improvement on the film processing is required. Although the ECD deposition has such a drawback, the advantages of ECD are evident as it is relevant for the large-area, low-cost, mass-production of thin film, which manifest important merits of spin-thermoelectric energy conversion.

#9. Reviewer's comments:

The units for magnetic fields and magnetization should be unified. The present version includes both the cgs Gauss and SI unit systems; the latter is better.

Our responses:

We greatly appreciate reviewer's considerate comments. As reviewer suggested, we unified all into the SI units in the revised manuscript.

#10. Reviewer's comments:

On page 5 of Supplementary Information, why did the authors determine the Curie temperature of Cr-PBA based on the magnetization curve at 10 Oe. Since 10 Oe is smaller than the coercive force, the magnetization curve is affected by the hysteresis. The Curie temperature should be determined by the Arrott plot method.

Our responses:

We greatly appreciate reviewer's valuable suggestion. As reviewer suggested, we performed the Arrott plot method within the mean-field theory to obtain the Curie temperature (T_c) of the studied Cr-PBA film. As shown in the Supplementary Fig. S3, the M^2 vs B/M curves exhibit good linearity with zero intercept at $T_c \sim 230$ K.

In response to the reviewer's comments, we added Arrott plot in Supplementary Fig. 3 in the revised manuscript. And we changed following sentence in the revised manuscript.

In page 6,

Original manuscript: The temperature-dependent magnetization curve shown in Supplementary Fig. 3 exhibits transition temperature of $T_c \sim 223$ K.

Revised manuscript: The Curie temperature of the developed Cr-PBA determined by Arrott plot method was $T_c \sim 230$ K (see Supplementary Fig. 3).

#11. Reviewer's comments:

On the same page, the authors state "Thus, the excitations of magnons with low wave vectors are more effective in our Cr-PBA film. This behavior indicates strong suppression of sub-thermal magnons will occur at high magnetic field,...", which is based on the difference in the temperature dependence of magnetization between Cr-PBA and Ni. However, this discussion is too rough and I do not understand why the authors compare the magnetization curve of Cr-PBA with that of Ni. Due to the spectral-dependent nature of SSE, it is difficult to predict its behavior only from static magnetic properties.

Our responses:

We greatly appreciate reviewer's valuable comments. We agree that our assumption was too rough. And other effects involve in the high-field suppression of LSSE signal. Thus, we removed our argument on the contribution of effective generation of spin wave on LSSE signal throughout the manuscript. We compared temperature dependent magnetization of Cr-PBA with Ni and YIG, because YIG is a standard magnetic insulator used for the study of spin Seebeck effect and Ni is one of the most standard ferromagnet, respectively.

In response to reviewer's comments we changed following sentences in the revised manuscript.

In Abstract, line 25

Original manuscript: "Prussian blue analogue, electrochemically deposited on Cr electrodes at room temperature show effective spin thermoelectricity associated with strong excitations of low energy spin waves."

Revised manuscript: "Prussian blue analogue, electrochemically deposited on Cr electrodes at room temperature show effective spin thermoelectricity"

And we removed following sentences and phrases.

Line 89-90 in the original manuscript

~~"In particular, temperature and field dependent characteristics of spin thermoelectricity exhibit strong excitations of low energy spin waves."~~

Line 174-175 in the original manuscript,

~~"The high field suppression of ΔV_{LSSE} can be attributed to the suppression of sub-thermal magnons"~~

Line 180-181 in the original manuscript,

~~"leading to strong suppression of V_{LSSE} at high magnetic field."~~

#12. Reviewer's comments:

The measurement temperature for the 3-omega method should be described. The error bars should be added to Supplementary Fig. 5.

Our responses:

In response to reviewer's comments, we added measurement temperature and further details for the thermal conductivity measurements including error bars.

In order to improve precision on the estimation of the cross-plane thermal conductivity, we performed 3-omega measurement again with the enlarged width of the top heater line from 20 μm to 100 μm . For anisotropic thermal conductivity measurements, the combination between heater wire width and the film thickness determine the sensitivity of the measurement to the in-plane and cross-plane thermal properties of the film. Choosing a heater width much larger than the film thickness, the measured temperature drop could be assumed to be sensitive mainly to the cross-plane heat dissipation [T. Borca-Tasciuc, Rev. Sci. Instrum., **72**, 4 (2001)]. If the wire width is smaller or comparable to the film thickness, the measured temperature signal is

influenced by both in-plane and cross-plane thermal conductivity of the film. Thus, the wider width of Au heater, which also has same geometry of LSSE device heater, allow us to reduce the error from in-plane heat dissipation and obtain more precise estimation for the cross-plane thermal conductivity of Cr-PBA. The obtained value of thermal conductivity measured at 100 K was 2.18 ± 0.01 W/mK. This value is slight higher than that we estimated before with 20 μm wide top heater line. Thus, we recalculated the temperature gradient applied on Cr-PBA film in Supplementary Table 1 and Supplementary Fig. 8 and Fig. 2d and Fig. 3 in the main text. We also added the error bar related with measurements of temperature drop. The standard deviation of ΔT between the sample and reference in heating currents of 50, 60, 70 mA was 0.0057 K, 0.0111 K, 0.0010 K, respectively. The estimated error propagations are represented in Supplementary Fig. 5.

In short, major changes we made in response to the reviewer's comments are as follows.

1. Addition of the discussion about the alternative electrodes for ECD deposition and high spin-charge conversion.
2. Addition of the geometry of the device, which produced $\Delta V_{\text{LSSE}}/\Delta T \sim 64.9 \pm 3.13$ $\mu\text{V}/\text{K}$ at 100 K.
3. Change the vector form throughout the manuscript.
4. Unified unit and expression for the magnetic field.
5. Addition of Supplementary Fig. 3 and description for the estimation of T_c using Arrott plot method.
6. Addition of Supplementary Fig. 4b for field-dependent magnetization with different applied magnetic field and its description
7. Addition of the discussion on the estimation of the temperature gradient applied on Cr-PBA
8. We re-estimated the thermal conductivity of Cr-PBA with improved geometry for the cross-plane κ and added error bar in the estimated value of thermal conductivity in Supplementary Fig. 5.
9. Addition of Supplementary Fig. 9 for the estimated spin Seebeck coefficient with various thicknesses
10. Addition of Supplementary Fig. 11a and the discussion about the temperature-dependent LSSE characteristics for different thickness of Cr-PBA.
11. Addition of Supplementary Fig. 11b for the measured temperature dependence of thermal conductivity of Cr-PBA.
12. Addition of Supplementary Fig. 12a and the discussion about the observed Nernst effect of the Cr layer in the device.
13. Addition of Supplementary Fig. 12b and the discussion about the high-field suppression of LSSE signal with different Cr-PBA thickness.
14. Addition of Supplementary Fig. 13 and the discussion about the high-field suppression of LSSE signal at various temperatures.
15. Removal of previous argument about the impact of effective excitation of low-energy spin waves on the strong high-field suppression of LSSE signal

Reviewer #2

Reviewer's comments:

The work by Oh et al. reports on the fabrication of vertical heterojunctions based on molecular Prussian blue analogue (PBA) compounds, their physical characterization and their possible use as a spin thermoelectric material.

The manuscript is clear, easy to read and the topic and results are of broad interest. I find it suitable for being published in Nature Communications once the following points are addressed:

Our responses:

We greatly appreciate reviewer's important comments and positive remarks. Below is the detailed discussions and corrections we made in response to reviewer's comments.

#1. Reviewer's comments:

How many different samples were measured for each of the performed experiments? Is there any dependence of the measured properties with the PBA thickness?

Our responses:

We have studied more than several samples with various thicknesses of Cr-PBA for the characterization of longitudinal spin Seebeck effect. When the device fabrication processes were done properly, all device exhibits robust LSSE voltage. The major results shown in the manuscript were obtained from one of the best samples for the ECD deposition time of 600 sec, which creates ~ 1.4 μm thick Cr-PBA. Supplementary Fig. 9 exhibits the measured spin Seebeck coefficients for different thickness of samples. Results display general tendency of the enhancement of the SSE coefficient with increasing thickness of Cr-PBA.

In response to reviewer's comments, we added Supplementary Fig. 9 and following sentences in the revised manuscript.

Line 157 (page 8) in the revised manuscript:

“Supplementary Fig. 9 shows the estimated S_{LSSE} for the devices with several different thicknesses of Cr-PBA. Results display general tendency of the enhancement of S_{LSSE} with increasing thickness of Cr-PBA.”

#2. Reviewer's comments:

In the Supplementary Figure 4, the field cool magnetization is performed with an external

magnetic field of 0.5 T (i.e., a considerable field). From the fitting of the data, the authors state “Fitting with $M(T) = 178 M_0(1 - aT^{3/2})$ provides the value of slope $a \sim 2.6 \times 10^{-4} \text{ K}^{-3/2}$. This value of slope is much higher than those observed in Ni ($7.5 \times 10^{-6} \text{ K}^{-3/2}$)³⁴ and YIG ($5.8 \times 10^{-5} \text{ K}^{-3/2}$)³². Thus, the excitations of magnons with low wave vectors are more effective in our Cr-PBA film, leading to strong suppression of VLSSE at high magnetic field.” How does the magnetization look like with a smaller applied field (like 10 Oe or 100 Oe)? Is the same slope obtained?

Our responses:

We appreciate the reviewer’s considerate comments. In response to reviewer’s comments, we added Supplementary Fig. 4b, which shows temperature dependent magnetization measured with applying various external magnetic fields. The slope of Bloch spin wave decreases with decreasing magnetic field. This could be due to the fact that smaller size of domain formed at low magnetic field would not be adequate for the effective excitation of low-energy and long-wavelength spin waves.

We initially attributed strong high-field suppression of LSSE signal to the strong suppression of magnon at high-magnetic field. However, after we checked the Nernst effect in our device at temperature above the Curie temperature of Cr-PBA, we noticed that linear suppression in V_{LSSE} with increasing field is largely originated from the Nernst effect of Cr electrode. Thus, we removed our argument on the contribution of effective generation of spin wave on LSSE signal throughout the manuscript as follows.

In Abstract, line 25

Original manuscript: “Prussian blue analogue, electrochemically deposited on Cr electrodes at room temperature show effective spin thermoelectricity associated with strong excitations of low energy spin waves.”

Revised manuscript: “Prussian blue analogue, electrochemically deposited on Cr electrodes at room temperature show effective spin thermoelectricity”

And we removed following sentences and phrases.

Line 89-90 in the original manuscript

~~“In particular, temperature and field dependent characteristics of spin thermoelectricity exhibit strong excitations of low energy spin waves.”~~

Line 174-175 in the original manuscript,

~~“The high field suppression of ΔV_{LSSE} can be attributed to the suppression of sub-thermal~~

magnons’

Line 180-181 in the original manuscript,

“~~leading to strong suppression of V_{LSSE} at high magnetic field.~~”

In short, major changes we made in response to the reviewer’s comments are as follows.

1. Addition of the discussion about the alternative electrodes for ECD deposition and high spin-charge conversion.
2. Addition of the geometry of the device, which produced $\Delta V_{\text{LSSE}}/\Delta T \sim 64.9 \pm 3.13 \mu\text{V/K}$ at 100 K.
3. Change the vector form throughout the manuscript.
4. Unified unit and expression for the magnetic field.
5. Addition of Supplementary Fig. 3 and description for the estimation of T_c using Arrott plot method.
6. Addition of Supplementary Fig. 4b for field-dependent magnetization with different applied magnetic field and its description
7. Addition of the discussion on the estimation of the temperature gradient applied on Cr-PBA
8. We re-estimated the thermal conductivity of Cr-PBA with improved geometry for the cross-plane κ and added error bar in the estimated value of thermal conductivity in Supplementary Fig. 5.
9. Addition of Supplementary Fig. 9 for the estimated spin Seebeck coefficient with various thicknesses
10. Addition of Supplementary Fig. 11a and the discussion about the temperature-dependent LSSE characteristics for different thickness of Cr-PBA.
11. Addition of Supplementary Fig. 11b for the measured temperature dependence of thermal conductivity of Cr-PBA.
12. Addition of Supplementary Fig. 12a and the discussion about the observed Nernst effect of the Cr layer in the device.
13. Addition of Supplementary Fig. 12b and the discussion about the high-field suppression of LSSE signal with different Cr-PBA thickness.
14. Addition of Supplementary Fig. 13 and the discussion about the high-field suppression of LSSE signal at various temperatures.
15. Removal of previous argument about the impact of effective excitation of low-energy spin waves on the strong high-field suppression of LSSE signal

Reviewers' Comments:

Reviewer #1:

Remarks to the Author:

The authors have carefully addressed my comments, and the manuscript is improved. I can recommend the publication of this paper in Nature Communications. However, before acceptance, the following additional comments should be addressed.

[Response to the previous comment #1]

Judging from the sample size information, the observed large spin Seebeck voltage, $\Delta V_{\text{LSSE}}/\Delta T$, is attributed to the significant difference between the thickness and length of the device. Since the S_{LSSE} value including the size information is very small, the magnitude of the spin Seebeck voltage cannot be an appealing point. The minimum revision to address this point is to remove "significantly large" on line 88 in the main text.

[Response to the previous comments #5 & #6]

Although the authors assume that "the Nernst effect is nearly temperature independent for metal", this has not been confirmed experimentally. Furthermore, in order to subtract the Nernst contribution, the magnitude of the temperature gradient at 300 K must be the same as that at the SSE measurement temperatures. However, since the thermal conductivity depends on the temperature, the temperature gradient value should also depend on the temperature (note that the thermal conductivity at 300 K is about twice as large as that at 100 K). Thus, I think that the accurate subtraction of the Nernst contribution is impossible, and it is difficult to conclude that the voltage slope in the strong magnetic field region after the subtraction is attributed to the suppression of SSE. The authors should discuss this point and revise the manuscript appropriately. Also, raw data before the subtraction should be provided for all the samples.

Reviewer #2:

Remarks to the Author:

The authors have addressed all my concerns, so, I find the manuscript suitable for being published in Nature Communications.

Responses to the reviewer's comments

Reviewer #1

Reviewer's Comments

The authors have carefully addressed my comments, and the manuscript is improved. I can recommend the publication of this paper in Nature Communications. However, before acceptance, the following additional comments should be addressed.

Our responses:

We greatly appreciate reviewer's positive remarks for the revised manuscript and important comments for improving the quality of our manuscript.

#1. Reviewer's comments:

Judging from the sample size information, the observed large spin Seebeck voltage, $\Delta V_{\text{LSSE}}/\Delta T$, is attributed to the significant difference between the thickness and length of the device. Since the S_{LSSE} value including the size information is very small, the magnitude of the spin Seebeck voltage cannot be an appealing point. The minimum revision to address this point is to remove "significantly large" on line 88 in the main text.

Our responses:

We agree with reviewer's opinion. We replaced "significantly large" with "robust" in line 89 in the revised manuscript.

#2. Reviewer's comments:

Although the authors assume that "the Nernst effect is nearly temperature independent for metal", this has not been confirmed experimentally. Furthermore, in order to subtract the Nernst contribution, the magnitude of the temperature gradient at 300 K must be the same as that at the SSE measurement temperatures. However, since the thermal conductivity depends on the temperature, the temperature gradient value should also depend on the temperature (note that the thermal conductivity at 300 K is about twice as large as that at 100 K). Thus, I think that the accurate subtraction of the Nernst contribution is impossible, and it is difficult to conclude that the voltage slope in the strong magnetic field region after the subtraction is attributed to the suppression of SSE. The authors should discuss this point and revise the manuscript appropriately. Also, raw data before the subtraction should be provided for all the samples.

Our responses :

We greatly appreciate the reviewer's considerate comments. We agree with reviewer's opinion

and suggestion.

In response to reviewer's comments, we made following changes in the revised manuscript.

1. Following phrase is added in the revised manuscript

In Line 185

“assuming that the Nernst effect in the Cr layer is temperature independent.”

2. Line 189 in the revised manuscript:

Original manuscript:

The high-field suppression of V_{LSSE} increases as increasing thickness of the Cr-PBA film, in consistent with previous works^{28,36} (Supplementary Fig. 12b).

Revised manuscript:

Previous studies with YIG attributed the high-field suppression of V_{LSSE} to the suppression of sub-thermal magnon^{28,36} (Supplementary Fig. 12b). However, the high-field suppression of the estimated V_{LSSE} in our studies could be due to the variation in the Nernst effect as the applied ΔT could be altered at different temperature (Supplementary Note 2).

3. Following sentences are added in the revised Supplementary Information.

In page 17,

“Results display high-field suppression ratio of the LSSE signal tends to increase with increasing thickness of Cr-PBA film, as observed in previous works^{10,11}. However, precise subtraction of the Nernst effect in our devices is unattainable task because the applied ΔT in the thin Cr layer will be largely temperature dependent. We estimated ΔT in each layer by using the Fourier's law. But, thermal conductivities in every layer are generally temperature dependent as in Cr-PBA, whose κ value at 300 K is nearly twice of the value at 100 K (Fig. S11b). Thus, the applied ΔT in the thin Cr layer is temperature dependent, so does the size of Nernst voltage. Therefore, actual high-field suppression in V_{LSSE} could be considerably different from the estimated $V_{\text{LSSE}}(B_y)$, shown in Fig. S12d.”

4. Raw data of Supplementary Fig. 12a and c are added in the revised Supplementary Information.